# A Fluoroponytailed NHC–Silver Complex Formed from Vinylimidazolium/AgNO_3_ under Aqueous–Ammoniacal Conditions [note 1]

**DOI:** 10.3390/molecules27134137

**Published:** 2022-06-28

**Authors:** Gabriel Partl, Marcus Rauter, Lukas Fliri, Thomas Gelbrich, Christoph Kreutz, Thomas Müller, Volker Kahlenberg, Sven Nerdinger, Herwig Schottenberger

**Affiliations:** 1Institute of General, Inorganic and Theoretical Chemistry, University of Innsbruck, CCB, Innrain 80-82, 6020 Innsbruck, Austria; gabriel.j.partl@gmail.com; 2Infineon Technologies AG, Siemensstrasse 2, 9500 Villach, Austria; marcus.rauter@infineon.com; 3Department of Bioproducts and Biosystems, Aalto University, P.O. Box 16300, 00076 Aalto, Finland; lukas.fliri@aalto.fi; 4Institute of Pharmacy, Leopold Franzens University of Innsbruck, Innrain 52c, 6020 Innsbruck, Austria; thomas.gelbrich@uibk.ac.at; 5Institute Organic Chemistry, University of Innsbruck, CCB, Innrain 80-82, 6020 Innsbruck, Austria; christoph.kreutz@uibk.ac.at (C.K.); thomas.mueller@uibk.ac.at (T.M.); 6Institute of Mineralogy & Petrography, University of Innsbruck, Innrain 52, 6020 Innsbruck, Austria; volker.kahlenberg@uibk.ac.at; 7Sandoz GmbH, Biochemiestr. 10, 6250 Kundl, Austria

**Keywords:** silver, *N*-heterocyclic carbene, fluoroponytail, crosslinker, cationic fluorosurfactant

## Abstract

3-(1*H*,1*H*,2*H*,2*H*-Perfluorooctyl)-1-vinylimidazolium chloride [2126844–17–3], a strong fluorosurfactant with remarkably high solubility in water, was expediently converted into the respective doubly NHC-complexed silver salt with nitrate as counter ion in quantitative yield. Due to its vinyl substituents, [bis(3-(1*H*,1*H*,2*H*,2*H*-perfluorooctyl)-1-vinylimidazol-2-ylidene)silver(I)] nitrate, **Ag(FNHC)_2_NO_3_**, represents a polymerizable *N*-heterocyclic carbene transfer reagent, thus potentially offering simple and robust access to coordination polymers with crosslinking metal bridges. The compound was characterized by infrared and NMR spectroscopy, mass spectrometry as well as elemental analysis, and supplemented by X-ray single-crystal structure determination. It crystallizes in the monoclinic crystal system in the space group *P*2_1_/c. With 173.3°, the geometry of the Ag-carbene bridge deviates slightly from linearity. The disordered perfluoroalkyl side chains exhibit a helical conformation.

## 1. Introduction

Evolved by nature [1], and eventually driven by refined fundamental understanding [2], *N*-Heterocyclic Carbene species (NHCs) have conquered a whole universe of practical applications in materials science [3] and especially in catalysis. As a pool of switchable, multifunctional, adaptable, or tunable ligands, they still gain growing relevance in the fields of organometallic chemistry [4] and organocatalysis [5]. Notably, each of the respective subtopics are covered and frequently updated by countless dedicated reviews. Among them, the following are particularly worth mentioning: olefin metathesis [6,7], C–H activation reactions [8], C–C coupling and olefin polymerizations [9]. Representing fundamental starting materials, Ag(I)-NHC complexes were originally used as carbene transfer agents [10] for the conversion into other metal NHC systems. Recently, however, they have emerged as powerful catalysts in their own right [11]. Unsurprisingly for coordination compounds of coinage metals [12,13,14,15], silver complexes (of course, including NHC derivatives) exhibit good biocompatibility and have opened new horizons in life sciences [16,17,18]. In particular, they garnered enormous attention in biological as well as pharmaceutical research, for example, as antimicrobial [19,20,21,22] and cytostatic agents [23,24,25,26,27]. The profiles of bioactivity depend on either lipophilicity [28] or fluorophilicity [29], which in turn are primarily contingent on the chain length of the nitrogen substituents. Differences in fluorophilicity for polyfluorinated NHC (F-NHC)-Ag complexes were recorded as a function of the type of polyfluorinated moiety. Compared to polyfluoroalkyl ponytails, fluorinated polyether chains resulted in much higher fluorophilicity. This was attributed to the conformational flexibility of the polyfluoropolyether chains that are able to shield fluorophobic counteranions against the perfluorinated environment, thereby minimizing fluorophobic interactions [29].

In the environmentally relevant context of polyfluorinated chain length in correlation with ecotoxicology, it should be emphasized that even short-chain fluorosurfactants belong to the group of persistent xenochemicals. As such, they too are to be eliminated, at least for non-essential uses, since growing evidence suggests that they are associated with similar adverse toxicological effects as long-chain per- and polyfluoroalkyl substances [30]. For example, in strict observance of process containment, chain backfluorinated NHC carbenes may prove indispensable in biphase fluorous catalysis [31]. As a counterexample, despite the excellent antifouling properties of polyurethanes concomitantly modified with perfluoroalkyl moieties and silver nanoparticles, such coatings are unlikely to have a commercial future for biomedical applications [32]. However, for some imidazolium-based F-NHC-precursors, the acute ecotoxicity profiles are occasionally even less detrimental than their hydrocarbon-based congeners [33].

In general, from the very onset of academic and industrial research on NHC complexes, studies of perfluoroalkyl- or perfluoralkylene containing NHC derivatives have been communicated regularly, but only sparsely with regard to this special area [34,35,36,37,38,39,40,41,42,43,44,45,46]. 

As a missing topic in our own contributions to fluorous imidazole chemistry [33,47], straightforwardly accessible NHC derivatives urged us to pursue such a project. In this context, **Ag(FNHC)_2_NO_3_** is presented as a first example and discussed in this communication. 

## 2. Results and Discussion

### 2.1. Synthetic Considerations

Since water was often considered the “natural enemy” of organometallic species [48], aqueous reaction conditions were carefully avoided in the early days of systematic NHC research [34]. Over time, however, synthetic methodologies surrounding NHCs became markedly less stringent [49,50]. By now, the use of water-soluble NHC-based catalysts has turned into well-established routine [48].

With the exception of persistent carbenes [51], the onset of NHC complexation equilibria usually requires the deprotonation of the conjugated acid of the NHC carbene, namely the corresponding azolium ions. Fortunately, even weak bases such as acetate, which often represent the counter-anionic constituent of starting metal salts or the organic heterocyclic salts, are completely sufficient [35]. Therefore, such pure ionic couples are susceptible to segregating NHC adducts when exposed to an adventive Lewis acid [52]. 

Synthesis design aspects relating to affordability and accessibility of starting materials, ease of workup, overall yields and scalability of intermediates as well as target compounds have to be taken into consideration, yet are often left to chance. Nevertheless, more efficient and less wasteful syntheses represent a basic prerequisite for a possible transfer into practical application. In the present case, intermolecular interactions and phase equilibria such as fluorophilic segregation and self-assemblies [53] are appreciably operative and facilitate the efficient isolation of the FNHC target compound (Figure 1). 

In the case of hydrophobic fluoroponytailed azolium-based ionic liquids, which belong to the same family of potential F-NHC precursors, it is worth mentioning that not only the counter ions but also the side chains accompanying the polyfluoroalkyl residue exhibit strong and tunable effects on the mutual solubility towards binary and ternary phases with aqueous and hydrofluoroether systems [54].

2-Vinylimidazolium salts still qualify as one of the best and most affordable starting compounds, in particular for the long-explored polymerizations of micelle-forming surfactants. However, additional functionalization is thereby limited [55].

The peculiar selective precipitation of **Ag^+^(FNHC)_2_** in the form of its nitrate salt rather than the chloride or mixed salt can be attributed to the lower hydration energy of NO_3_**^−^** compared to Cl**^−^** [56]. The so-called dehydration phenomenon explains the difference in selectivity for different ions with similar net charges and hydrated radii by the extent to which dehydration occurs: the higher the hydration energy, the more difficult for the ion to undergo dehydration [57]. Conclusively, upon the onset of precipitation of the highly fluorous micellar domains [58] of **Ag^+^(FNHC)_2_**, it is plausible that the underlying interfacial partition equilibria of NO_3_**^−^** and Cl**^−^** eventually result in the formation of less hydrophilic **Ag(FNHC)_2_NO_3_**, and, due to its aforementioned higher charge density, Cl**^−^** remains in the aqueous phase. Furthermore, the Donnan (charge)-exclusion mechanism exerts a synergistic effect on the electrostatic interaction between different ions and the fixed charge on any ion exchange resin or related nanofiltration systems [59].

Since NHC ligands can be decorated with a plethora of functional groups [60], they are excellent building blocks for the construction of coordination architectures with desired properties. In particular, oligomers [61] and complexes featuring additional covalent polymerizability [22,25,36] are obvious candidates that promote materials research towards the next level of applicative relevance. For example, the remediation of polyfluoroalkyl substances (PFAS) by polymer ionic exchange resins (ionic fluorogels) is considered superior to the usual sorption on charcoal. Furthermore, the synergistic combination of fluorous and electrostatic interactions results in unequaled affinity, high capacity, and rapid sorption of a variety of PFASs from contaminated water [62]. However, fighting fire with fire still harbors issues of optimization.

The acquisition of fundamental knowledge on such compounds bridges the gap between sustainable and fluorous chemistry: while these materials may pose potential concerns, it is also true that fluorous chemistry will not disappear any time soon. Therefore, new technologies to deal with it are required [63,64].

### 2.2. Crystallography

The asymmetric unit of **Ag(FNHC)_2_NO_3_** contains one formula unit (Figure 1a). Both the Ag–C bond lengths of the carbene bridge of 2.075(6) Å and 2.083(6) Å and the corresponding C–Ag–C bond angle of 173.3(2)°, deviating slightly from linearity, lie within the typical range associated with carbene-Ag-carbene structures (see Appendix A). The conformation of the two chains units in the cation can be characterized in terms of a sequence of one N–C–C–C (*t*_1_) and six consecutive C–C–C–C torsion angles (*t*_2_ to *t*_6_). In both chains, *t*_1_ is *gauche* and all of *t*_2_ to *t*_6_ are *trans*. The terminal torsion angles *t*_4_ to *t*_6_ characterize the geometry of the respective C_6_F_13_ fluoroalkyl tail and show a twist of 14.5° and 9.8° (average deviation from 180°), indicating helical conformations. Due to the size of the fluorine atom, a fluoroalkyl chain (C_n_F_2n+1_) generally exhibits greater stiffness than a corresponding alkyl chain, resulting in a loss of *gauche*/*trans* freedom [65]. This typically leads to helical rather than planar zig-zag conformations, as found in perfluorohexane and other examples [33,66]. The C_6_F_13_ tail of each chain of **Ag(FNHC)_2_NO_3_** is disordered over two conformations, each representing the same principal geometry (occupancy ratios between disorder fragments 0.665:0.335 and 0.64:0.36). Additionally, the cation is disordered over two orientations (occupancy ratio 0.51:0.49). Within the crystal structure, the perfluoroalkyl chains of neighboring cations are stacked in a parallel/antiparallel fashion, resulting in a sequence of alternating perfluoroalkyl ponytail and polar domains parallel to the *bc* plane (Figure 1b). The molecular packing is dominated by a multitude of C–F⋯F–C contacts within these molecular stacks.

### 2.3. Spectroscopy and Supplementary Analyses

In addition to X-ray single-crystal structure determination, the obtained silver carbene complex was complementarily analyzed by elemental analysis, IR and NMR spectroscopy, and high-resolution mass spectrometry. While it was difficult to assess the success of the reaction via FT-IR owing to strong C-F vibrations that dominate the educt as well as the product spectrum (Appendix A), the ^1^H NMR spectrum proves the formation of a singular carbene species, as evidenced by the complete disappearance of the C2 proton and shifts in the heterocyclic resonances (Appendix A). Additionally conducted ^13^C (Appendix A), ^19^F (Appendix A) and ^19^F–^13^C HSQC NMR experiments (Appendix A) showed six peaks for the perfluorohexyl moieties, revealing that the complex sustains a chemically equivalent conformation and thus exists as **Ag^+^(FNHC)_2_** in solution. 

The elemental composition, according to combustion elemental analysis, gave satisfactory results for hydrogen and nitrogen, but underestimated the carbon content. This is presumably attributable to the reluctant combustion behavior of the perfluorinated moieties. 

In the high-resolution mass spectrum (Appendix A), the m/z of the molecule ion peak is in agreement with the molecular mass of **Ag^+^(FNHC)_2_**. The other prominent peak at approximately 441 Da corresponds to the 3-(1*H*,1*H*,2*H*,2*H*-perfluoroctyl)-1-vinylimidazolium ion, likely formed through in situ protonation of the carbene complex.

## 3. Materials and Methods

The fluoroponytailed starting reactant 3-(1*H*,1*H*,2*H*,2*H*-perfluoroctyl)-1-vinylimidazolium chloride, [2126844–17–3], was prepared following a published procedure [47]. All other chemicals and solvents were purchased from commercial sources, e.g., from Sigma Aldrich/Merck and used without further purification.

NMR spectra were recorded on a Bruker Avance DPX 300 MHz spectrometer or on a 700 MHz Avance 4 Neo spectrometer, equipped with a TCI Prodigy probe. The ^19^F spectra were externally referenced to CFCl_3_. The ^19^F-^13^C HSQC experiment was acquired with the following parameters: TD 1024 × 64; spectral widths: 60 ppm (^19^F) × 18 ppm (^13^C); ^1^*J* CF 240 Hz, number of scans: 16. Two ^19^F decoupled ^13^C spectra were recorded by placing the ^19^F carrier frequency at −120 ppm and −80 ppm. IR spectra were obtained with a Bruker ALPHA Platinum FT-ATR instrument. Mass spectrometric data were measured on a Thermo Finnigan Q Exactive Orbitrap spectrometer equipped with a HESI source. The infusion experiments were performed using MeOH as solvent, the spray voltage was 3.3 kV in the positive ion mode and the capillary temperature was 320 °C. Elemental analyses were conductedat the Laboratory for Microanalysis Services, Technical University of Vienna, Währingerstr. 42, 1090 Vienna, Austria; tracking number 522/0765 (https://chemie.univie.ac.at/en/services/service-facilities/lab-for-microanalysis-services, accessed on 26 June 2022).

Diffraction intensity data were recorded with an Oxford Diffraction Xcalibur Ruby Gemini diffractometer using Mo*Kα* (*λ* = 0.71073 Å) radiation. The crystal structure was solved by Direct Methods and refined by full-matrix least-squares techniques [67,68].

**Ag(FNHC)_2_NO_3_**, (C_26_H_18_AgF_26_N_4_)^+^(NO_3_)^−^: empirical formula C_26_H_18_AgF_26_N_5_O_3_; formula weight 1050.32; *T* = 173(2) K; monoclinic space group *P*2_1_/*c*; *Z* = 4; unit cell parameters *a* = 16.4683(9) Å, *b* = 19.2664(14) Å, *c* = 11.4836(6) Å, α = γ = 90°, β = 92.371(5)°, *V* = 3640.5(4) Å^3^; 22,435 reflections collected; 6638 independent reflections (*R*_int_ = 0.0355); *R*1 [*I* > 2σ(*I*)] = 0.0720; *wR*2 (all data) = 0.2132.

**CCDC 2163728** contains the supplementary crystallographic data for this paper. These data can be obtained free of charge from the Cambridge Crystallographic Data Centre via www.ccdc.cam.ac.uk/data_request/cif, accessed on 26 June 2022.

### Preparation of [Bis(3-(1H,1H,2H,2H-perfluoroctyl)-1-vinylimidazol-2-ylidene)silver(I)] Nitrate, **Ag(FNHC)_2_NO_3_**

3-(1*H*,1*H*,2*H*,2*H*-Perfluoroctyl)-1-vinylimidazolium chloride (4.8 g; 10 mmol), [2126844–17–3], was dissolved in water (10 mL), followed by addition of 25% ammonia solution (7.5 g; 110 mmol) under stirring. Subsequently, silver nitrate (0.85 g; 5 mmol) dissolved in water (10 mL) was added dropwise under stirring. The reaction mixture was protected from light using aluminum foil and agitated at room temperature for one hour. Afterwards, the precipitated product was filtered off, washed portion-wise with a total of 50 mL of cold water, and dried in vacuo for 24 h. Then, 5.2 g (4.95 mmol, 99% of theoretical yield) of a white, powdery solid were isolated. Single crystals suitable for X-ray structure determination were grown from acetone by slow evaporation of the solvent.

Mp: 138 °C (under decomposition). FT-IR (ATR, neat) *ν* = 3108, 1650, 1429, 1345, 1231, 1183, 1141, 1124, 1075, 959, 909, 832, 808, 770, 745, 734, 699, 650, 568, 527, 448, 427 cm^−1^. ^1^H NMR (300 MHz, acetone-*d*_6_) *δ* = 7.97 (d, *J* = 2.1 Hz, 2H), 7.82 (d, *J* = 2.1 Hz, 2H), 7.59 (dd, *J* = 15.7, 8.9 Hz, 2H), 5.77 (dd, *J* = 15.7, 2.1 Hz, 2H), 5.15 (dd, *J* = 8.9, 2.1 Hz, 2H), 4.82 (t, *J* = 7.1 Hz, 4H), 3.06 (tt, *J* = 19.2 Hz, 7.1 Hz, 4H) ppm. ^13^C NMR (176 MHz, acetone-*d*_6_) *δ* = 184.2 (2C), 135.1 (2C), 124.0 (2C), 119.1 (2C), 118.3 (m, 2C–CF2), 117.2 (m, 2C–CF3), 111.1 (m, 2C–CF2), 110.8 (m, 2C–CF2), 110.3 (m, 2C–CF2), 108.5 (m, 2C–CF2), 104.8 (2C), 45.13–45.03 (m, 2C), 32.8 (t, *J* = 20.9 Hz, 2C) ppm. ^19^F NMR (659 MHz, acetone-*d*_6_) *δ* = −81.77 (t, *J* = 10.0 Hz, 6F), −114.08 (p, *J* = 17.6 Hz, 4F), −122.45 (4F), −123.48 (4F), −123.97 (4F), −126.84 (td, *J* = 15.2, 7.1 Hz, 4F) ppm. HRMS (ESI+) [M + H]^+^ for C_26_H_18_F_26_N_4_Ag^1+^
*calcd.*: m/z = 987.0162, *found*: m/z = 987.0115. Anal. *Calcd.* for C_26_H_18_AgF_26_N_5_O_3_: C, 29.73; H, 1.73; N, 6.67. *Found*: C, 28.61; H, 1.55; N, 6.36.

## 4. Conclusions and Outlook

F-chains combine two characteristics that are commonly considered antinomic: they are extremely hydrophobic and in addition have a pronounced lipophobic character [69,70]. Finally, as mentioned previously, under strict observation of process containment, leach-proof fluorosurfactants of the polyelectrolyte family may be qualified to belong to fluoropolymers of low concern [71] and may possibly serve as a “fluorous remedy” of polluting PFAS which are still deployed as production aids in the manufacture of fluoropolymers.

When functionalized in a smart way, all types of NHC complexes offer extremely inviting starting points for materials and life sciences [72]. In this respect, research dealing with the subfamily of FNHC complexes too will continue to produce further innovations in the future. Under these aspects, we developed a facile, robust and scalable procedure for fluoroponytailed, crosslinkable NHC complexes synthesized from easily accessible starting materials. By leaving the choice between chloride or nitrate as a counterion, **Ag(FNHC)_2_NO_3_** appeared as a first polymerizable paradigm, which was thoroughly characterized. In conclusion, the synergism of ammonium hydroxide solutions as a complexing and solubilizing auxiliary for silver salts with otherwise low solubility in aqueous solvent mixtures, along with the basic action of excess ammonia, makes it suitable for deprotonation equilibria of imidazolium salts into carbenes. This can be exploited as a well-known peculiarity for the straightforward synthesis of further NHC-transfer agents. Therefore, we expressly invite interested researchers to use this toolbox and gain new insights, be it in the field of ligand design, carbene transfer chemistry, specialty polymers or catalysis.

## Data Availability

Marcus Rauter, BSc; diploma thesis, Innsbruck 2020.

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
