# Peer review of "A Fluoroponytailed NHC–Silver Complex Formed from Vinylimidazolium/AgNO3 under Aqueous–Ammoniacal Conditions†"

_molecules, 2022, doi:10.3390/molecules27134137_

Round 1

Reviewer 1 Report

The authors of the present manuscript describe the synthesis of N-heterocyclic carbenes containing a fluoro ponytailed moiety.

The synthetic discussion is well introduced, as well as the rationale for the formation of the NO3 derivative instead of the chloride one.

The molecular structure of the complex is well discussed, but the discussion of the infrared spectra or the NMR is absent.   9F NMR would be particularly relevant. The compound is insufficiently characterizes; microanalysis is absent and HRMS is also not described.

Several typos are persistent throughout the manuscript, and must be corrected.

I do not understand why there is a section for materials and methods and another one for synthesis. This is probably a typo too, and should be corrected.

Also, the discussion section is not a discussion, is a set of considerations on the potential of the complexes and as such is better suited for a intro section. Similarly, the conclusions section, new considerations on the characteristics of the F-chains and NHCs that contained them are described, but the conclusion of the work itself is very vague.

Summarizing, I think that the work is interesting, but the manuscript requires further work and reorganization.

Author Response

Dear Madame or Sir,

            Re: Molecules-1736720, A Fluoroponytailed NHC- Silver Complex formed from Vinylimidazolium /AgNO3 under Aqueous-Ammoniacal Conditions

please compare attached word-file regarding my answer to your comments.

Yours sincerely,

Sven Nerdinger

Reviewer 2 Report

The manuscript Molecules-1736720 by Nerdinger and Schottenberger
et al. reports on the synthesis of bis N-heterocyclic carbene Ag(I) complex containing fluoroponytailed group on one nitrogen center of the NHC-ligand while the other nitrogen center contains a vinyl substituent. The deprotonation of the azolium salt occurs under aqueous ammoniac conditions, subsequent treatment with AgNO3 provides the target bis-carbene silver complex. The X-ray molecular structure of the silver salt is reported and confirms the formation of the desired molecule. The synthesis of such type of metal carbene complexes containing polyfluorinated chain on the NHC-ligand remains rare and might show interesting properties as suggested by the authors. However there is some work on carbene metal complexes that needs to be cited in the revised version before this paper gets accepted. See below:

1-Refs related to NHC-metal complexes to be cited: Anti-Cancer Agents Med. Chem. 2021, 21, 938; J. Organomet. Chem. 2020, 921, 121364.; Inorg. Chem. 2019, 58, 2930; Inorganics 2017, 5, 58.

2-P4 line 166. The synthesis of the azolium salt with fluoroponytailed groups is described in ref. 44 (J. Fluor. Chem. 2021, 249, 109839) and not ref. 43. Please correct.

3-The novel bis-carbene silver salt was characterized by NMR spectroscopy and supplemented by X-ray diffraction on a single crystal of the compound. It is necessary to obtain also the mass spectroscopy of the isolated product or its elemental analysis.

Therefore I recommend its publication in Molecules after the above issues are satisfactorily addressed.

Author Response

(The authors gave the same response as above.)

Reviewer 3 Report

The manuscript presented for review presents the synthesis of new fluoroponytailed NHC- Silver Complex that may be of interest in material science. I recommend the manuscript for publication in Molecules after addressing a number of recommendations.

1) The elemental analysis or HRMS data for characterization of new silver complex should be included in experimental part.

2) Scheme 1. It would be desirable for clarity to include to scheme footnote some synthetic details (reagents quantity, reactions conditions etc).

3) The references on the other application of fluorinated NHC complexes should be added (for example: Eur. J. Org. Chem. 2019, 2019, 1016-1020; Mendeleev Commun. 2018, 28, 609-611).

Author Response

(The authors gave the same response as above.)

Round 2

Reviewer 1 Report

The paper has improved with respect to data discussion and more insights on the compound are provided.

Still, I have some concerns that I would like to point out.

-       The language, at times, is still very confusing…. For instance “pursue realization” not sure what it means or “procedurial disrespect” regarding the development of synthesis using water.

-       The authors discuss the synthesis of the azolium salt, but the azolium salt is already reported, so I don’t understand the need to highlight the drawbacks of a procedure that is not to be used, since the detailed a synthesis is already described.

-       Line 130: dissolved state should be replace for “in solution”

-       Point 3 is highlighted as a discussion, but I still don’t understand why these considerations, which are not a discussion of results, require a different section.

-       My point on the conclusion section remains. There is no conclusion regarding the synthesis and characterization, only general remarks regarding the field and not the work developed in the paper.  

-       On the 2D NMR 19F - 13C HSQC , the 13 C does not correspond with the 13C spectrum indicated in the same document. Can the authors clarify?

-       For the elemental analysis, since the values for H N and C are discussed in the text, should be indicated as well in the S.I.   

Author Response

Please see attached word document regarding the answers.

Sincerely yours,

Sven Nerdinger
